# Dynamics of Antibody and T Cell Immunity against SARS-CoV-2 Variants of Concern and the Impact of Booster Vaccinations in Previously Infected and Infection-Naïve Individuals

**DOI:** 10.3390/vaccines10122132

**Published:** 2022-12-13

**Authors:** Michel R. Faas, Willem A. Mak, Hilde Y. Markus, Ellen M. van der Zwan, Marijke van der Vliet, Johannes G. M. Koeleman, David S. Y. Ong

**Affiliations:** 1Department of Medical Microbiology and Infection Control, Franciscus Gasthuis & Vlietland, Kleiweg 500, 3045 PM Rotterdam, The Netherlands; 2Department of Clinical Chemistry, Franciscus Gasthuis & Vlietland, Kleiweg 500, 3045 PM Rotterdam, The Netherlands; 3Department of Epidemiology, Julius Center for Health Sciences and Primary Care, University Medical Center Utrecht, Universiteitsweg 100, 3584 GC Utrecht, The Netherlands

**Keywords:** COVID-19, SARS-CoV-2, immunity, vaccination, T cell, antibody, variants

## Abstract

Despite previous coronavirus disease 2019 (COVID-19) vaccinations and severe acute respiratory syndrome coronavirus 2 (SARS-CoV-2) infections, SARS-CoV-2 still causes a substantial number of infections due to the waning of immunity and the emergence of new variants. Here, we assessed the SARS-CoV-2 spike subunit 1 (S1)-specific T cell responses, anti-SARS-CoV-2 receptor-binding domain (RBD) IgG serum concentrations, and the neutralizing activity of serum antibodies before and one, four, and seven months after the BNT162b2 or mRNA-1273 booster vaccination in a cohort of previously infected and infection-naïve healthcare workers (HCWs). Additionally, we assessed T cell responses against the spike protein of the SARS-CoV-2 Delta, Omicron BA.1 and BA.2 variants of concern (VOC). We found that S1-specific T cell responses, anti-RBD IgG concentrations, and neutralizing activity significantly increased one month after booster vaccination. Four months after booster vaccination, T cell and antibody responses significantly decreased but levels remained steady thereafter until seven months after booster vaccination. After a similar number of vaccinations, previously infected individuals had significantly higher S1-specific T cell, anti-RBD IgG, and neutralizing IgG responses than infection-naïve HCWs. Strikingly, we observed overall cross-reactive T cell responses against different SARS-CoV-2 VOC in both previously infected and infection-naïve HCWs. In summary, COVID-19 booster vaccinations induce strong T cell and neutralizing antibody responses and the presence of T cell responses against SARS-CoV-2 VOC suggest that vaccine-induced T cell immunity offers cross-reactive protection against different VOC.

## 1. Introduction

More than two years after coronavirus disease 2019 (COVID-19) was declared a pandemic [1], severe acute respiratory syndrome coronavirus 2 (SARS-CoV-2) still causes a substantial number of infections despite many individuals having been previously vaccinated against COVID-19 or having had a SARS-CoV-2 infection that builds immunity against the virus [2,3,4]. The presence of neutralizing antibodies is generally considered a key correlate of immune protection from SARS-CoV-2 infection [5,6]. These antibodies bind the receptor-binding domain (RBD) of the spike protein, thereby preventing the virus from entering human cells [7]. In addition to the humoral compartment, it is now widely accepted that T cells also play a pivotal role in controlling SARS-CoV-2 infection. For example, lymphopenia is a determinant for worse clinical outcomes after SARS-CoV-2 infection and memory T cell responses are maintained against multiple SARS-CoV-2 epitopes [8]. Accordingly, a considerable number of SARS-CoV-2 immunity studies revealed that SARS-CoV-2 infection and COVID-19 vaccination induce the formation of neutralizing anti-spike antibodies and robust T cell responses against a wide range of viral epitopes [9,10,11,12,13]. Although these immune responses were still detectable up to one year post-immunization, a significant decrease was observed within the first months following immunization [9,14,15,16]. This observation could at least partly explain why a substantial number of immunized individuals are (re)infected with the virus [17,18,19].

Besides the waning of SARS-CoV-2 immunity, the substantial incidence of new SARS-CoV-2 infections after earlier infection or vaccinations can potentially be explained by new emerging SARS-CoV-2 variants of concern (VOC), including the most recent VOC Delta (B.1.617.2 lineage) and Omicron (B.1.1.529 lineage) [3]. These SARS-CoV-2 VOC involve mutations in the spike protein and multiple studies reported that spike-specific antibodies partially lost their neutralizing capabilities against new SARS-CoV-2 VOC [20,21,22]. Remarkably, comparable results were observed within the same variant as the Omicron subvariants BA.4 and BA.5 escaped from neutralizing antibodies that were formed after Omicron BA.1 or BA.2 infection [23,24,25,26].

The present study aims to describe the long-term kinetics of SARS-CoV-2 specific humoral and T cell responses after primary and booster vaccinations in previously SARS-CoV-2-infected individuals and compare these to infection-naïve vaccinated individuals. In addition, we determined whether prior infection and vaccination induce cross-reactive T cell responses against the spike protein of the SARS-CoV-2 Delta and Omicron BA.1 and BA.2 VOC.

## 2. Materials and Methods

### 2.1. Study Design

This study cohort consisted of previously infected healthcare workers (HCWs) who tested SARS-CoV-2 reverse transcription-quantitative polymerase chain reaction (RT-qPCR) positive between March 2020 and March 2021, recently infected HCWs who tested RT-qPCR positive between December 2021 and May 2022, and infection-naïve HCWs who never tested SARS-CoV-2 RT-qPCR positive during the study period. All participating HCWs were affiliated with our hospital. For previously infected HCWs, SARS-CoV-2-specific T cell and antibody responses were measured at the following time points: June 2020 (only antibodies) and June 2021 (as part of our previous studies) [9,27], November 2021 (t0), December 2021 (t1), March 2022 (t2), and June 2022 (t3). The recently infected and infection-naïve HCWs were included in March 2022 (t2) and June 2022 (t3). This study received approval from the Medical Research Ethical Committee United (protocol number R20.030).

### 2.2. PBMC and Serum Isolation

The method of peripheral blood mononuclear cells (PBMC) and serum isolation was described in detail in our previous publication [9]. In short, the whole blood of the HCWs was obtained via venipuncture, serum was isolated from the whole blood, and PBMCs were isolated via Ficoll–Paque density gradient separation. The PBMC concentration was determined with an automated white blood cell counter.

### 2.3. SARS-CoV-2 S1 and N Interferon-Gamma (IFN-γ) ELIspot

T cell responses against the SARS-CoV-2 spike subunit 1 (S1) and nucleocapsid protein (N) were assessed by either the T-SPOT^®^ Discovery SARS-CoV-2 or T-SPOT^®^.COVID (both Oxford Immunotec, Abingdon, UK), which are identical assays that included similar S1 and N peptide pools. The ELIspot assay was performed exclusively with materials from the kit according to the manufacturer’s instructions. We refer to our previous publication for the methodological aspects of this ELIspot [9].

### 2.4. SARS-CoV-2 Variant IFN-γ ELIspot

We used an in-house-developed SARS-CoV-2 variant IFN-γ ELISpot. On day 1, polyvinylidene fluoride membranes precoated with a monoclonal anti-IFN-γ antibody (mAb 1-D1K, Mabtech, Stockholm, Sweden) were washed three times with PBS and conditioned with AIM-V (AIM-V^®^ + AlbuMAX^®^ (BSA); Gibco, Carlsbad, CA, USA) for 30 min at room temperature. The following stimulations were added in a volume of 50 µL per well: AIM-V as a negative control, anti-CD3 as a positive control (1:1000, mAb CD3-2, Mabtech), and the following SARS-CoV-2 spike peptide pools consisting mainly of 15-mer sequences with 11 amino acids overlaps were added to a final 1.0 µg/mL concentration: Omicron BA.1 mutation and corresponding wild-type (WT) pool (PepTivator^®^ SARS-CoV-2 Prot_S B.1.1.529/BA.1, Miltenyi Biotec, Bergisch Gladbach, Germany), BA.2 mutation and corresponding WT pool (PepTivator^®^ SARS-CoV-2 Prot_S B.1.1.529/BA.2, Miltenyi Biotec), and Delta mutation and corresponding WT pool (PepTivator^®^ SARS-CoV-2 Prot_S B.1.617.2, Miltenyi Biotec). An amount of 2.5 × 10^5^ PBMCs in 50 µL AIM-V was added to each well, whereafter the microtiter plate was incubated for 16–20 h at 37 °C with 5% CO_2_ in a humidified atmosphere. On day 2, the PBMCs were washed off the plate with PBS, and 100 µL alkaline phosphatase-conjugated antibody (1:200, 7-B6-1-ALP, Mabtech) was added to the wells and was incubated for two hours at room temperature. Subsequently, the microtiter plate was again washed with PBS and 100 µL substrate (BCIP-NBT-plus; Mabtech) was added to the wells and was incubated at room temperature for 7–12 min, whereafter the reaction was stopped with demineralized water.

### 2.5. ELIspot Image Processing and Spot Quantification

As previously described [28], the images of the ELISpot membranes were captured with the DX1 USB microscope (Veho) at a resolution of 1280 × 960, using Plugable Digital Viewer v3.1.07 (Plugable) software. Using FIJI v2.1.0 software [29], the images were circularly cropped and converted to 32 bits (black and white), whereafter the spot intensity threshold was set at 75. Next, the images were converted to an 8-bit mask and a particle analysis that only included spots ≥ 5 pixels was performed. Samples were excluded if the positive control condition resulted in <100 spots. The number of spots in the negative control condition was subtracted from the number of spots counted in peptide-stimulated conditions within the same sample.

### 2.6. SARS-CoV-2 Anti-RBD IgG Quantitative ELISA

Serum anti-SARS-CoV-2-RBD IgG concentrations were assessed with a quantitative enzyme-linked immunosorbent assay (ELISA) (Beijing Wantai Biological Pharmacy Enterprise, Beijing, China) that was performed according to the manufacturer’s guidelines in a fully automated microplate processor and analyzer (ETI-MAX, Diasorin, Saluggia, Italy). The 32 Wantai-units per mL (U/mL) standard, part of the ELISA kit, was serially diluted to create a range from 32.0 U/mL to 1.0 U/mL. These standard concentrations were used to create a calibration line, whereafter serum IgG concentrations were calculated. As described by the manufacturer, the calculated IgG concentrations were converted from U/mL to international units per mL (IU/mL) using the conversion factor of 5.4. Serum samples were diluted to up to 1:500 in order to fit in the concentration range of the calibration line. HRP-conjugate was incubated at 37 °C for 30 min and tetramethylbenzidine (TMB) substrate was incubated at 37 °C for 15 min. The reaction was stopped with sulfuric acid and the absorbance was measured at 450 nm.

### 2.7. SARS-CoV-2 IgG Surrogate Virus Neutralization Assay

The surrogate virus neutralization test (sVNT) kit (Genscript Biotech) was used to measure anti-SARS-CoV-2-RBD neutralizing activity [27]. The sVNT was performed according to the manufacturer’s guidelines in a fully automated microplate analyzer (ETI-MAX, Diasorin). Serum samples were diluted by up to 1:500, whereafter HRP-RBD was incubated at 37 °C for 30 min, 100 µL pre-incubated mix was added and incubated at 37 °C for 15 min, and TMB substrate was incubated at 21 °C for 15 min. The reaction was halted with sulfuric acid and the absorbance was measured at 450 nm. The neutralizing anti-RBD-antibody quantity was expressed as a neutralizing activity percentage, which was acquired using the following calculation: neutralizing activity (%) = (1 − OD value of sample/ OD value of negative control) × 100.

### 2.8. SARS-CoV-2 Anti-N IgG Qualitative CMIA

A chemiluminescent microparticle immunoassay (CMIA) (SARS-CoV-2 IgG, Abbott, Chicago, IL, USA) was performed according to the manufacturer’s instructions using an automated immunoassay analyzer (Architect, Abbott) to qualitatively detect anti-SARS-CoV-2-N IgG in serum samples. The binding of antibodies to the N antigen-coated paramagnetic microparticles was measured in relative light units (RLU), whereafter each sample’s RLU was compared to the RLU of the calibrator. RLU indices were calculated and an index of ≥1.4 was considered positive.

### 2.9. Statistical Analysis

All data obtained in this study were expressed as median with interquartile range (IQR) and statistical analyses were performed with GraphPad Prism v9 (GraphPad Software). Wilcoxon signed-rank test was performed to compare paired datasets, whereas the Mann–Whitney U test was applied to compare two independent data sets. The Kruskal–Wallis test with Dunn’s multiple comparison test was used to compare two groups within three or more independent groups. Spearman’s rank correlation coefficients were calculated to assess associations between groups and were interpreted as negligible (r < 0.1), weak (r = 0.11–0.39), moderate (r = 0.40–0.69), strong (r = 0.70–0.89), or very strong (r = 0.90–1.00) [30]. All statistical tests were performed at a two-tailed α- level of 0.05.

## 3. Results

The HCW study cohort has been previously described with results up to June 2021 [9]. In this study, 35 previously infected HCWs continued to participate in this follow-up study from November 2021 (Figure 1). Of these participants, thirteen, ten, nine, and four had received BNT162b2, mRNA-1273, Ad26.CoV2-S, and ChAdOx1-S as part of their primary vaccination series, respectively. In between t0 and t1, the HCWs received an mRNA booster vaccination. Two new HCW groups were introduced in March 2022: recently infected HCWs, who were (re)infected with SARS-CoV-2 between December 2021 and March 2022, and infection-naïve HCWs. Of the recently infected HCWs, three, two, and two participants had received BNT162b2, mRNA-1273, and Ad26.CoV-S as their primary vaccination series, respectively, whereas one HCW had received a combination of BNT162b2 and mRNA-1273. Of the infection-naïve HCWs, twenty-one, one, and two individuals had received BNT162b2, mRNA-1273, and Ad26.CoV-S, respectively.

### 3.1. SARS-CoV-2 Specific T Cell and Antibody Responses Five Months Post-Primary COVID-19 Vaccinations

In the previous study, the SARS-CoV-2 specific T cell and antibody levels of previously infected HCWs were assessed two weeks after the completion of the primary vaccination series (i.e., one or two vaccinations depending on vaccine brand) in June 2021 [9]. In order to investigate the waning of immune responses in the following five months, we assessed the changes in these levels in 32 of 35 HCWs five months post-vaccination in November 2021. Compared to two weeks after vaccination, the S1-specific T cell responses were significantly lower at five months post-vaccination (*p* = 0.0037) (Figure 2A). In more detail, the median S1-specific T cells decreased from 126 (IQR 89–228) to 74 (IQR 46–132) spot-forming cells (SFCs) per 10^6^ PBMCs. In contrast, N-specific T cell responses remained comparable between the two time points, with a median of 16 (IQR 8–32) and 18 (IQR 4–36) N-specific SFCs per 10^6^ PBMCs, respectively. Moreover, we observed significantly lower anti-RBD IgG serum concentrations and the lower neutralizing activity of these antibodies (both *p* < 0.0001) five months post-vaccination (Figure 2B,C). Median anti-RBD IgG concentrations significantly decreased from 3415 (IQR 2071–6631) to 907.5 (IQR 449.5–1400) IU/mL and the median neutralizing activity of antibodies decreased from 67.4% (IQR 47.1–90.3%) to 28.8% (IQR 16.1–40.0%) (both *p* < 0.0001).

### 3.2. The Effect of Booster Vaccination in Previously Infected Individuals

Next, we assessed the T cell and antibody responses of previously infected HCWs three weeks after receiving a COVID-19 booster vaccination. We observed a significant increase in S1-specific T cells three weeks after booster vaccination (*p* < 0.0001), which is illustrated by the median SFCs per 10^6^ PBMCs that increased from 82 (IQR 50–255) to 234 (IQR 147–500) (Figure 3A). Similarly, the median serum anti-RBD IgG concentrations increased from 1085 (IQR 477.2–1516) to 2540 (IQR 1601–4574) IU/mL and the median neutralizing activity increased from 29.2% (IQR 16.6–41.8%) to 63.0% (IQR 48.1–85.2%) after booster vaccination (both *p* < 0.0001) (Figure 3B,C). Furthermore, there were non-significant weak associations between the number of S1-specific T cells and anti-RBD IgG concentrations at t0 (r = 0.3398, *p* = 0.0894) and t1 (r = 0.3211, *p* = 0.1097) (Figure 3D). In contrast, significant strong associations were found between the neutralizing activity of serum antibodies and serum anti-RBD IgG concentrations at both t0 (r = 0.7642, *p* < 0.0001) and t1 (r = 0.8277, *p* < 0.0001) (Figure 3E).

### 3.3. Durability of SARS-CoV-2-Specific T Cell and Antibody Responses after Booster Vaccination in Previously Infected Individuals

To determine the durability of T cell and antibody responses induced by booster vaccination, we assessed SARS-CoV-2-specific T cell and antibody responses four and seven months after booster vaccination in previously infected HCWs. We observed that the S1-specific T cell response was significantly decreased (*p* < 0.0001) four months post-booster vaccination from a median of 328 (IQR 160–1036) to 104 (IQR 64–336) SFCs per 10^6^ PBMCs (Figure 4A). Similarly, the anti-RBD IgG concentrations and neutralizing activity decreased (*p* < 0.0001 and *p* < 0.0003, respectively) from a median of 3510 (IQR 2023–4788) to 1332 (IQR 705.8–2581) IU/mL and from 74.1% (IQR 45.9–88.3%) to 37.1% (IQR 22.8–73.5%), respectively (Figure 4B,C). In contrast to the first four months after booster vaccination, we observed that the number of S1-specific T cells and anti-RBD IgG concentrations remained similar after the subsequent three months (i.e., t3) (Figure 4D,E). Moreover, the anti-RBD IgG concentrations at four and seven months after booster vaccination were significantly higher compared to the concentration at five months after the primary vaccination series (*p* = 0.0202 and *p* = 0.0003, respectively). However, S1-specific T cell responses were not significantly different between these time points.

**Figure 3 vaccines-10-02132-f003:**
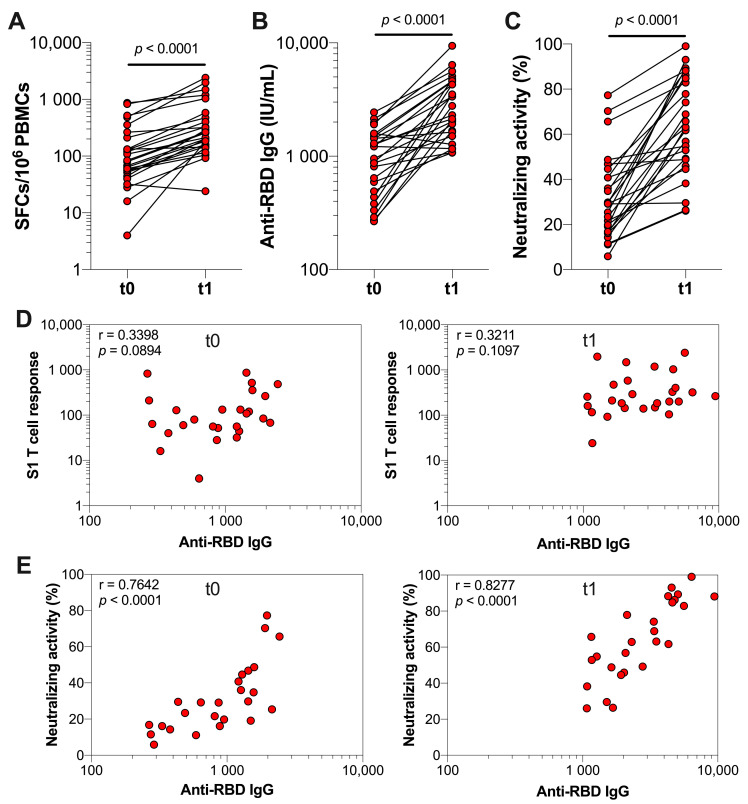
SARS-CoV-2-specific T cell and antibody responses of previously infected HCWs before and after booster vaccination. Immune responses of previously infected HCWs (*n* = 26) were assessed at median 1 (IQR 1–7) day before (t0) and 27 (IQR 24–29) days after (t1) receiving the BNT162b2 or mRNA-1273 booster vaccination. Each data point represents an individual HCW. (**A**) T cell responses against SARS-CoV-2 S1 and N. (**B**) Serum anti-RBD IgG concentrations. (**C**) Neutralizing activity of serum antibodies against SARS-CoV-2. (**D**) Association between SARS-CoV-2 S1-specific T cell responses and anti-RBD IgG concentrations at t0 and t1. (**E**) Association between the neutralizing activities and anti-RBD IgG concentrations at t0 and t1. Statistical significance between paired data was assessed with a Wilcoxon test (**A**–**C**) and associations were assessed by Spearman’s rank correlation (**D**,**E**).

### 3.4. Comparison of Immune Responses between Previously Infected, Recently Infected and Infection-Naïve Individuals

At four months after booster vaccination (t2), 14 recently infected HCWs and 24 infection-naïve HCWs were introduced to the study cohort. Flow cytometric analysis demonstrated similar absolute concentrations of total leukocytes, total lymphocytes, B cells (CD19+), NK cells (CD3−, CD56+), T cell (CD3+), CD4+ T cells, and CD8+ T cells between previously infected, recently infected, and infection-naïve HCWs (Appendix A). We observed that previously and recently infected individuals exhibited significantly higher S1-specific T cell responses than infection-naïve HCWs at t2 (*p* = 0.0002 and *p* = 0.0038, respectively), whereas these responses were similar between previously and recently infected HCWs (Figure 5A). Serum anti-RBD IgG concentrations were significantly higher in previously and recently infected HCWs compared with infection-naïve HCWs (*p* = 0.0406 and *p* = 0.0003, respectively) (Figure 5B). Moreover, we observed higher neutralizing activity in previously and recently infected HCWs than in infection-naïve HCWs (*p* = 0.0121 and *p* = 0.0001, respectively) (Figure 5C). Median anti-RBD IgG concentrations and neutralizing activity for previously infected, recently infected, and infection-naïve HCWs were 1332 (IQR 804–2581), 2201 (IQR 1143–4101), and 859 (IQR 643–1141) IU/mL, while neutralizing activity was 38.3% (IQR 23.0–69.2%), 57.9% (IQR 31.7–90.1%), and 20.3% (IQR 11.3–30.1%), respectively.

Seven months post-booster vaccination, we observed significantly higher numbers of S1-specific T cells in previously infected HCWs (*p* = 0.0138) compared to infection-naïve HCWs at t3, whereas there were no differences between the recently infected and infection-naïve HCWs (Figure 5D). Median S1-reactive T cells for previously infected, recently infected, and infection-naïve HCWs were 68 (IQR 32–196), 48 (IQR 24–96), and 28 (IQR 12–66). Previously and recently infected HCWs showed significantly higher N-specific T cell responses than infection-naïve HCWs at t2 and t3 (*p* <0.0001 for all, data not shown). Furthermore, significantly higher anti-RBD IgG concentrations were observed in previously and recently infected HCWs compared to infection-naïve HCWs (*p* = 0.0258 and *p* < 0.0001, respectively) (Figure 5E) as median values for previously, recently, and infection-naïve HCWs were 1606 (IQR 1097–2130), 4239 (IQR 1767–6350), and 1002 (IQR 673–1390) IU/mL.

The longitudinal analysis of the three HCW groups revealed no significant differences in S1 and N-specific T cell responses and anti-RBD IgG concentrations between t2 and t3 (Appendix A). Moreover, significant strong associations were found between the neutralizing activity of serum antibodies and anti-RBD IgG concentrations at t2 for previously infected (r = 0.7678, *p* < 0.0001), recently infected (r = 0.7902, *p* = 0.0033), and infection-naïve HCWs (r = 0.8105, *p* < 0.0001) (Appendix A). Furthermore, S1-specific T cells and anti-RBD IgG concentrations at t3 were only significantly associated in the infection-naïve HCW group, presenting a moderate association (r = 0.6055, *p* = 0.0036).

Finally, we determined the dynamics of serum anti-N IgG over a two-year period in previously infected HCWs in June 2020, June 2021, and March 2022 (t3). In June 2020, i.e., one to two months after infection, 90% of previously infected HCWs tested positive for anti-N IgG presence in serum (Figure 5F). In the following two years, anti-N IgG was detected in 19% of HCWs in June 2021 and 5% in March 2022, suggesting that the assessment of anti-N IgG demonstrates a low sensitivity in regard to the detection of older infections.

### 3.5. SARS-CoV-2 VOC Spike-Specific T Cell Responses in Previously Infected, Recently Infected and Infection-Naïve Individuals

To determine whether prior infection and vaccination induce T cell responses against the spike protein of SARS-CoV-2 VOC, the PBMCs of previously infected, recently infected, and infection-naïve were stimulated with an Omicron BA.1, Omicron BA.2, and Delta spike peptide pool at t3. Since these peptide pools were composed of solely variant-specific mutation-containing peptides, we compared the reactivity against these peptides to the reactivity against SARS-CoV-2 wild-type (WT) spike peptide pools, which contained congruent peptides but without the mutations. We observed that all HCW groups showed comparable T cell responses between the mutation and WT peptide pools of all tested variants (Figure 6A–C).

Furthermore, there were significantly higher Omicron BA.1 spike-specific T cell responses in previously and recently infected HCWs than in infection-naïve HCWs (*p* = 0.0121 and *p* = 0.0450, respectively) (Figure 6D). Omicron BA.2 spike-specific T cell responses were significantly higher in previously infected HCWs compared to infection-naïve HCWs (*p* = 0.0001). In contrast, no significant differences were observed in Delta spike-specific T cell responses between the HCW groups. Median SFCs per 10^6^ PBMCs for previously infected, recently infected, and infection-naïve HCWs were 40 (IQR 12–100), 32 (IQR 12–76), and 16 (IQR 4–22) after Omicron BA.1 spike peptide pool stimulation, 52 (IQR 18–94), 24 (IQR 8–52), and 8 (IQR 2–20) after Omicron BA.2 spike peptide pool stimulation and 8 (IQR 0–24), 8 (IQR 0–24), and 4 (IQR 0–8) after Delta spike peptide pool stimulation, respectively.

## 4. Discussion

In this study, we showed that SARS-CoV-2 S1-specific T cells, anti-RBD IgG, and neutralization antibody activity waned significantly five months post-primary vaccinations. Booster vaccination significantly increased these immune responses, which was followed by a significant decrease again four months thereafter and subsequently remained steadier afterward. Our findings suggest that T cell responses induced by COVID-19 vaccination, which did not contain spike proteins of VOC, are cross-reactive against the spike protein of SARS-CoV-2 Delta and Omicron BA.1 and BA.2 VOC. Additionally, individuals who were infected with the ancestral virus strain in 2020 exhibited stronger T cell responses against Omicron BA.1 and BA.2 mutation-specific regions of the spike protein than infection-naïve individuals.

As expected, both T cell and antibody responses significantly increased after booster vaccination. Similar to the immune response dynamics after primary COVID-19 vaccinations [9,31,32,33,34], we observed that T cell and antibody responses significantly waned during the first months after booster vaccination, which was also observed in other studies [35]. Since serum antibody concentrations wane after vaccination, neutralizing IgA and IgG antibodies on mucosal surfaces are possibly also decreased, which may at least partially explain why vaccine breakthrough infections have been abundantly reported [7,36,37].

Interestingly, we found that T cell and anti-RBD IgG antibody responses were higher in previously infected HCWs compared to infection-naïve HCWs at four and seven months after booster vaccination. These results are in line with findings from other studies [34,35]. These studies also reported that anti-RBD IgG levels waned at a slower rate after the booster vaccination compared to after the primary vaccination in infection-naïve individuals. Similarly, we showed that serum anti-RBD IgG concentrations were higher four and seven months post-booster vaccination than five months post-primary vaccinations. Thus, all these observations together suggest that the number of antigen exposures is associated with the durability of SARS-CoV-2 spike-specific T cell and antibody responses.

In line with our findings, several previous studies also reported that spike-specific T cells induced by prior SARS-CoV-2 infection or COVID-19 vaccination can still recognize the spike proteins of SARS-CoV-2 variants, including Omicron [22,38,39,40,41]. The high-resolution analysis of spike peptide specificities mapping peptide-specific CD4+ T cell responses in prior-infected or vaccinated individuals showed that most reactive peptides were conserved in BA.4 and BA.5, suggesting that existing spike-specific T cells can recognize the spike protein of BA.4 and BA.5 [42]. Therefore, in contrast to neutralizing antibodies, there seems to be an overall preservation of T cell immunity against the spike protein. This indicates that T cells may play an important role as a second line of defense if the VOC escapes from the neutralizing antibodies.

Remarkably, anti-N IgG antibodies were detected in 90% of HCWs several weeks after testing SARS-CoV-2 RT-qPCR positive, which decreased to 19% and 5% after 12 months and 21 months, respectively. This is in line with other studies showing that anti-N IgG in serum was undetectable in most previously infected individuals [43,44,45]. In contrast, anti-RBD IgG seems more durable as one study reported a half-life of 212 days for anti-spike-RBD IgG versus 100 days for anti-N IgG [44]. The presence of anti-N IgG is, therefore, likely indicative of a more recent SARS-CoV-2 infection.

This study is one of the first to quantitatively describe both cellular and humoral SARS-CoV-2-specific responses at different time points before and after receiving an mRNA booster vaccination in a prospective cohort for more than two years ongoing. In addition, we determined whether vaccine-induced T cell responses were cross-reactive against SARS-CoV-2 VOC. This study also has several limitations to consider. First, the sample size in each group was limited. However, we were still able to statistically compare the different groups and find significant differences. Second, the ELISpot assay does not characterize reactive T cells, such as the differentiation between CD4+/CD8+ subsets, and only detects IFN-γ-secreting effector T cells. Flow cytometry-based assays, such as an activation-induced marker (AIM) assay or multimer-based assays, would be required to detect both effector and memory T cells, distinguish between T cell subsets, and determine other cytokine responses [46,47]. Third, we did not assess T cell reactivity against the current dominant Omicron subvariant BA.5 since BA.5 peptides were not yet commercially available during this study.

In conclusion, SARS-CoV-2-specific T cell and IgG responses waned after primary vaccinations over several months, were significantly boosted again after an mRNA booster vaccination, but waned again during the following four months post-booster vaccination, only to remain at steadier levels thereafter. Furthermore, both old and recent previous infections contributed to higher immune responses. In contrast to neutralizing antibodies that are variant-specific, vaccine-induced T cell responses are cross-reactive against various VOC. Future research should confirm whether vaccine-induced T cell responses are also cross-reactive against the Omicron subvariant BA.5 and new emerging variants.

## Figures and Tables

**Figure 1 vaccines-10-02132-f001:**
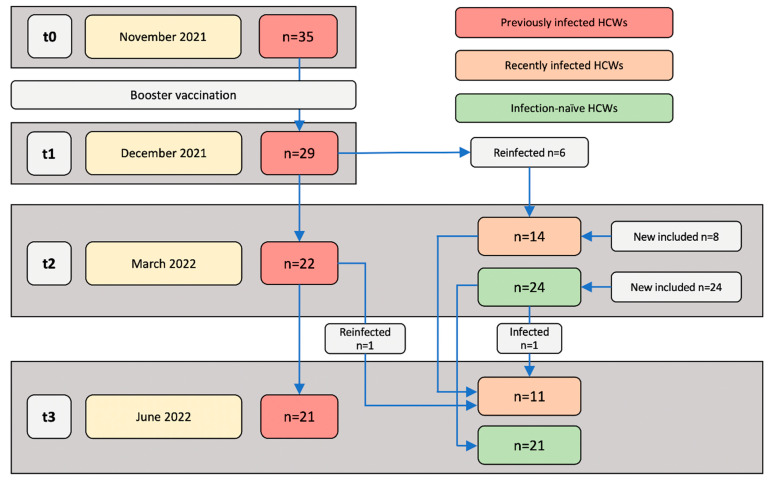
Study populations and sampling time points. Blood samples were collected and immune responses were assessed at the following four time points: November 2021 (t0; approximately two weeks before receiving a booster vaccination), December 2021 (t1; one month post-booster vaccination), March 2022 (t2; four months post-booster vaccination), and June 2022 (t3; seven months post-booster vaccination).

**Figure 2 vaccines-10-02132-f002:**
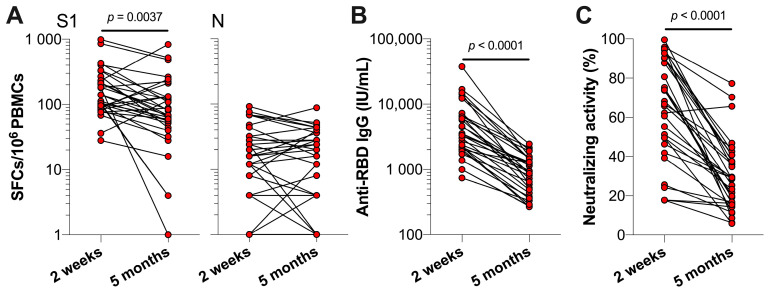
SARS-CoV-2-specific T cell and antibody responses of previously infected HCWs two weeks and five months post-primary vaccinations. Previously infected HCWs (*n* = 32) are represented by individual data points. (**A**) T cell responses against SARS-CoV-2 S1 and N, (**B**) serum anti-RBD IgG concentrations, and (**C**) the neutralizing activity of serum antibodies against SARS-CoV-2 at two weeks and five months (i.e., t0) after primary vaccination series. Statistical significance was assessed with a Wilcoxon test.

**Figure 4 vaccines-10-02132-f004:**
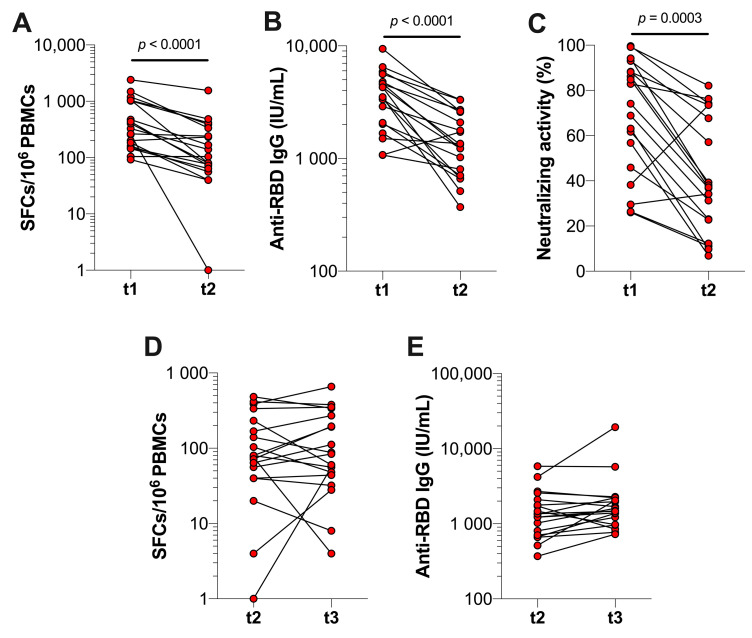
SARS-CoV-2-specific T cell and antibody responses of previously infected HCWs at t1, t2, and t3. Immune responses of previously infected HCWs were assessed median 27 (IQR 22–28) days (t1), 125 (IQR 118–126) days (t2), and 221 (IQR 204–213) days (t3) after receiving BNT162b2 or mRNA-1273 booster vaccination. (**A**) SARS-CoV-2 S1-specific T cell responses, (**B**) serum anti-RBD IgG concentrations, and (**C**) neutralizing activity against SARS-CoV-2 at t1 and t2 (*n* = 19). (**D**) SARS-CoV-2 S1-specific and (**E**) serum anti-RBD IgG concentrations at t2 and t3 (*n* = 19). Statistical significance was assessed with a Wilcoxon test.

**Figure 5 vaccines-10-02132-f005:**
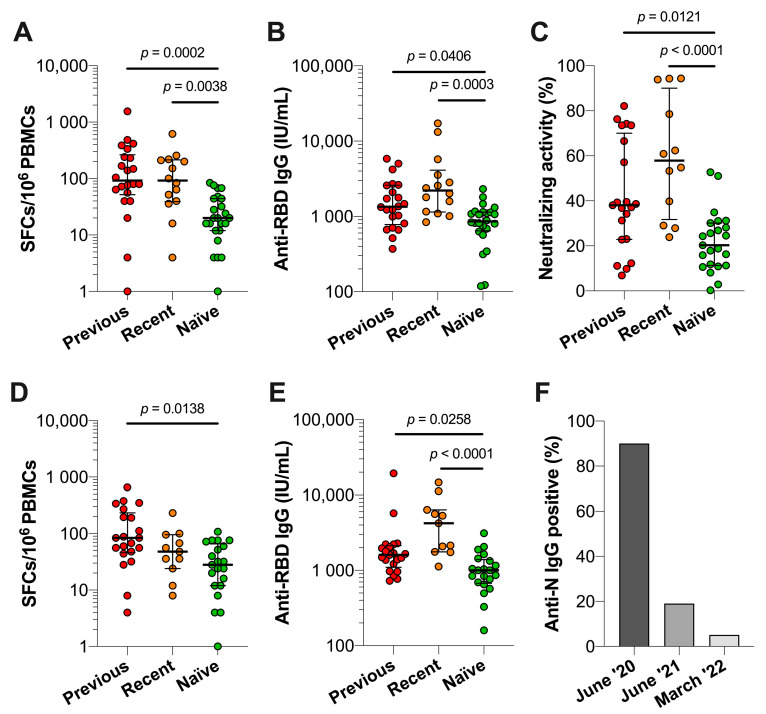
SARS-CoV-2-specific T cell and antibody responses of previously infected, recently infected, and infection-naïve HCWs. Immune responses were assessed at t2 (median 126 (IQR 122–131) days after receiving BNT162b2 or mRNA-1273 booster vaccination) and t3 (median 209 (IQR 204–212) days after receiving BNT162b2 or mRNA-1273 booster vaccination). (**A**) SARS-CoV-2 S1-specific T cell responses at t2 of previously infected (*n* = 22), recently infected (*n* = 14), and infection-naïve HCWs (*n* = 24). (**B**) Anti-RBD serum IgG concentrations at t2 of previously infected (*n* = 22), recently infected (*n* = 14), and infection-naïve HCWs (*n* = 24). (**C**) Neutralizing activity against SARS-CoV-2 RBD at t2 of previously infected (*n* = 21), recently infected (*n* = 12), and infection-naïve HCWs (*n* = 23). (**D**) T cell responses against SARS-CoV-2 S1 and N at t3 between previously infected (*n* = 21), recently infected (*n* = 11), and infection-naïve HCWs (*n* = 21). (**E**) Anti-RBD serum IgG concentrations at t3 between previously infected (*n* = 21), recently infected (*n* = 11), and infection-naïve HCWs (*n* = 21). Indicated is the median with IQR and statistical significance was assessed with a Kruskal–Wallis test with Dunn’s multiple comparison test. (**F**) Percentages of serum anti-N IgG-positive previously infected HCWs (*n* = 21) assessed at June 2020, June 2021, and March 2022 (t2).

**Figure 6 vaccines-10-02132-f006:**
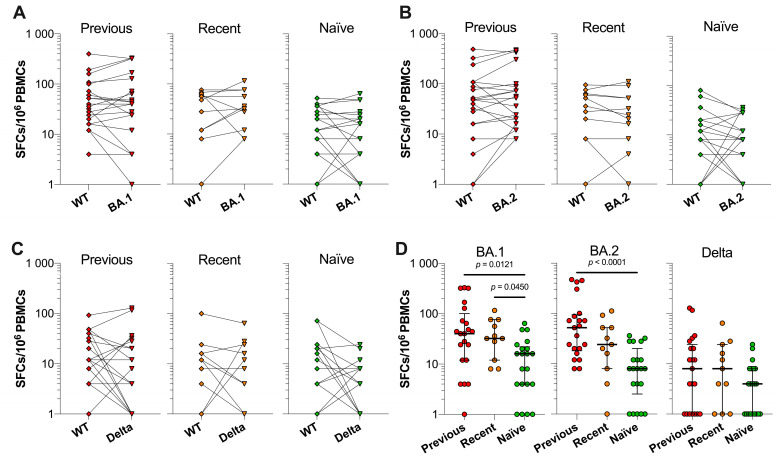
T cell responses against wild-type spike and VOC-mutated spike proteins. At t3, the PBMCs of previously infected (*n* = 21), recently infected (*n* = 11), and infection-naïve (*n* = 21) HCWs were stimulated with SARS-CoV-2 Omicron BA.1, Omicron BA.2, and Delta mutation spike peptide pools and corresponding SARS-CoV-2 wildtype (WT) spike peptide pools. (**A**) T cell responses against the SARS-CoV-2 Omicron BA.1 spike and corresponding WT spike peptide pools. (**B**) T cell responses against the SARS-CoV-2 Omicron BA.2 spike and corresponding WT spike peptide pools. (**C**) T cell responses against the SARS-CoV-2 Delta spike and corresponding WT spike peptide pools. Statistical significance was assessed with a Mann–Whitney test. (**D**) T cell responses against SARS-CoV-2 Omicron BA.1, Omicron BA.2, and Delta spike mutation peptide pools between previously infected, recently infected, and infection-naive HCWs. Indicated is the median with IQR and statistical significance was assessed through a Kruskal–Wallis test with Dunn’s multiple comparison test.

## Data Availability

Not applicable.

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
