# Peer review of "Dynamics of Antibody and T Cell Immunity against SARS-CoV-2 Variants of Concern and the Impact of Booster Vaccinations in Previously Infected and Infection-Naïve Individuals"

_vaccines, 2022, doi:10.3390/vaccines10122132_

Round 1

Reviewer 1 Report

Faas and colleagues undertook an important study assessing the dynamics of Ab and T cell immunity against Delta and Omicron variants/subvariants and the impact of vaccination and/or natural infection on these responses. Using well characterized protocols, they were able to reiterate previously reported findings that neutralizing Ab responses wane significantly over a period of few months. Surprisingly while Spike specific T cell responses waned significantly between 2 weeks and 5 months, N specific T cell responses remained comparable. Importantly vaccination boosters appeared to elicit more durable Spike specific T cell responses compared to primary series and in particular, participants that had been previously infected appeared to have more pronounced responses. While this study focused solely on the Delta and omicron variants, it does set a precedence as to the importance of boosters in providing protection against neutralization resistance strains such as Omicron.  This work is highly relevant and the authors should be commended on contributing such valuable input to our ever evolving understanding of the COVID-19 pandemic and strategies that can be employed to combat this pathogen.

Minor comments

-       Typo at line 189. Random number 9

-       Authors never specified primary vaccination series. May be relevant to the responses elicited by the mRNA booster? Perhaps specify in the study design. It might be interesting to note that irrespective of the primary series/type of vaccination, mRNA boosters potentially are able to provide durable T cell immunity

Reviewer 2 Report

Correct first sentence of abstract is very length.

Modified this sentence

Additionally, we assessed T cell responses against the spike protein of SARS-CoV-variants of concern (VOC) Delta and Omicron BA. and BA.2. We found that S1-specific T cell responses, anti-RBD IgG concentrations, and neutralizing activity were significantly increased one month post booster vaccination.

Carefully check throughout and remove grammatical mistakes.\

Clearly write methodology.

What's the new in this paper write clearly. Show in abstract and conclusion. 
